# Cultural Landscape Reproduction of Typical Religious Architecture in Qingjiangpu Based on Scene Theory

Wei Mao [1,2], Shuai Hong [1,2], Tengfei Chai [1,2], Junchao Shen [1,2] and Jie Shen [1,2,3,*]

1 School of Geographic Sciences, Nanjing Normal University, Nanjing 210023, China
2 Key Laboratory of Virtual Geographic Environment, Ministry of Education, Nanjing 210023, China
3 Jiangsu Center for Collaborative Innovation in Geographical Information Resource Development and Application, Nanjing 210023, China
* Correspondence: shenjie@njnu.edu.cn

**Abstract:** Scenes are important carriers of cultural expression. Cultural landscapes reveal specific cultural connotations through various scenes, and people understand and give things cultural connotations through scenes. In recent years, new techniques for visualizing cultural landscape heritage have been made possible by the advent of mapping and geographic information technology. The Beijing-Hangzhou Grand Canal's culture is a "living" cultural legacy. As one of the key links in the canal's cultural chain, Qingjiangpu is crucial to reproducing its cultural landscape. This paper first discusses the relationship between scene theory and the cultural landscape. Starting from the five elements of scene theory, through the collection of online text data and the corresponding data obtained from questionnaire research, the paper analyzed the scene constructed by the cultural landscape and the urban spirituality embodied by the scene. Through the deep excavation of cultural landscape and its historical context, the theoretical framework of "node-neighbor-city" cultural landscape reproduction is proposed. Taking the ancient city of Qingjiangpu as an example, the cultural landscape has been reproduced at different scales and in different dimensions through various technical means. This study can provide a theoretical basis and practical reference for the research of cultural landscape reproduction.

**Keywords:** scene theory; cultural landscape; religious architecture; Qingjiangpu; reproduction

## 1. Introduction

In 1992, UNESCO included cultural landscapes in its list of intangible cultural heritage, defining them as the product of interactions between humans and their environment within a particular geographic region [1]. Cultural landscapes' induction into the World Heritage list has opened up fresh avenues for the interpretation of local values and culture. However, on the one hand, due to natural changes and human activities, cultural landscapes have unavoidably experienced erosion and devastation [2]. On the other hand, people have put forward new requirements for the heritage of global cultural landscapes, science education, and seminars and exchanges. How to better preserve, inherit and disseminate cultural landscape heritage is an important historical mission [3]. In this situation, combining information visualization technology with traditional cultural heritage display and protection, the cultural landscape heritage is represented in a more vivid and interactive way through a series of technical means. These can include high-resolution digital image acquisition, data processing, three-dimensional modeling, virtual reality, and the intersection of multidisciplinary knowledge such as sociology, human geography, historical geography, and architecture. The reproduction of cultural landscape scenes can reproduce the cultural connotation conveyed by the cultural landscape, which is of great significance to the development of the city and the inheritance of culture.

With the rapid development of science and technology, a large number of information technology tools have been applied to the collection, processing, and expression of cultural

heritage landscape data, providing strong theoretical and technical support for modeling and its application research [4,5]. The digitization of cultural heritage landscapes contains three main areas of research, namely, digital modeling, 3D digital representation, and human-computer interaction. Digital modeling refers to the process of digitizing cultural heritage landscapes. The main techniques include 3D scanning [6,7], photogrammetry [8–11] and traditional geometric modeling, and image-based modeling [12–14], etc. 3D digital representation [15] is an intelligent display of cultural heritage landscape combining reality and imagination, and it is a key part of the digital cultural heritage display and interactive system. Virtual reality and augmented reality are often used for the digital representation of cultural heritage [16,17]. Facing the demand for 3D display of cultural heritage, scholars combine text, audio, video, and story retrieval to carry out the digitization of cultural heritage landscapes. Due to the special characteristics of information about cultural heritage, human-computer interaction is a crucial issue in virtual experiences. It is crucial in virtual experiences of cultural heritage. Relevant research has realized human-computer interaction [18] through narrative and immersive experiences in the form of multimedia museums [19,20] and 360° video [21]. The development of digital cultural heritage in depth and breadth offers more possibilities for digitizing cultural heritage landscapes on a technical level. However, at the same time, it is also very important to analyze and understand the elements, themes, and styles of the cultural heritage landscape. There are few papers on how to scientifically select and determine the theme content of urban cultural landscape recreation, and how to show the historical and cultural connotation of the city through cultural heritage landscape digitization. Therefore, this study attempts to propose a method to improve this situation.

Scene theory was proposed by the New Chicago School led by Terry Clark and Daniel Silver, who emphasized the concept of "scene" to explain the cultural content in space, and to analyze and find the cultural style of a place. Scene theory believes that different scenes formed by the combination of various urban cultural facilities with people and activities contain different scene values, and these attract people with the same cultural value orientation to living and participate, and finally can promote the renewal of the area and the development of the city [22].

Scene theory has been introduced by many scholars to conduct relevant research, mainly in urban development, cultural space creation, and cultural consumption promotion, but there are few studies on cultural landscape reproduction. Given this, this paper firstly analyses the cultural landscape scenes and the cultural values shown in the scenes from the perspective of the "scene theory" of the New Chicago School, deeply digs into the cultural landscape and the historical context of its formation, and puts forward the "node-neighborhood-city" theoretical framework of cultural landscape reproduction. Based on its unique historical background, the ancient city of Qingjiangpu is used as an example to collect and organize multiple sources of information on the history, geography, and religious architecture of the ancient city of Qingjiangpu in its main historical periods. The unique cultural landscape of religious architecture of "Southern boats and Northern horses, five religions architectures gathering" is reproduced from three different scales: macroscopic, mesoscopic, and microscopic. Through the above study, the religious architecture culture with Chinese characteristics, based on the background of canal culture, will be inherited and carried forward, providing a reference for the research related to cultural landscape reproduction.

## 2. Scene Theory and Cultural Landscape

### 2.1. Scene Theory

The term "scene" is widely used in many disciplines, including film and television, communication, and sociology. Initially, the term "scene" was used to refer to images in drama, film, and television. Later, the field of communication science introduced the concept of the "media scene" to emphasize a behavioral and psychological environment created by media messages. Robert Scoble and Shel Israel first combined scenes with the Internet, and in their book *Age of context: Mobile, sensors, data and the future of privacy*, they built mobile devices, social media, big data, sensors, and location systems together as the

five technological forces of scenes, and predicted that the Internet would enter the era of scenes in 25 years [23]. As a result, the term "scene" has become familiar with the popularity of the Internet and is widely used in the field of marketing. In the field of sociology, Terry Clark of the New Chicago School was the first to use the term "scene". According to Terry Clark, the scene is a holistic concept that includes five components: (1) neighborhood; (2) physical structures; (3) persons labeled by race, class, gender, education, etc.; (4) the specific combinations of these and activities; and (5) the value bred in the scene [22,24], as shown in Figure 1. As a new paradigm of urban research of the New Chicago School, the scene theory extends the study of urban space from the level of natural and social attributes to the level of consumption practice of regional culture. After their research on big cities, such as New York, Los Angeles, Chicago, and Paris, they found that different combinations of urban amenities will form different "scenes", and different "scenes" contain specific cultural values, thus forming specific spaces and cultural connotations [25].

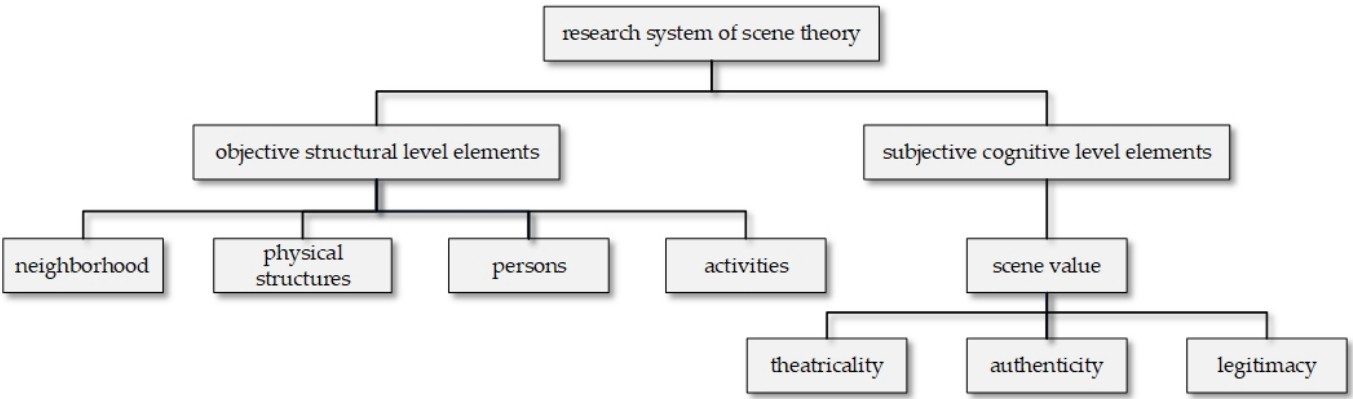

**Figure 1.** The research system of scene theory [24].

*2.2. Cultural Landscape*

Landscapes can be divided into natural landscapes and cultural landscapes, and cultural landscapes have a long history of study in disciplines such as human geography, ecology, and archaeology. The German geographer, Humboldt, conducted landscape studies at the beginning of the 19th century, arguing that landscape should be the central issue of geography and exploring the process of changing from a primitive natural landscape to a cultural landscape. Sauer, an American human geographer in the 20th century [26] studied cultural landscapes and believed that cultural landscapes are a form of human activity attached to natural landscapes. The study of cultural landscapes is helpful to understand the spatial differences of human culture, and even reflects the natural historical background that forms regional culture. The concept of cultural landscape was incorporated into the World Heritage List at the 16th session of the UNESCO World Heritage Committee, held in Santa Fe, USA, in December 1992. As a result, World Heritage Sites are divided into natural heritage, cultural heritage, natural and cultural composite heritage, and cultural landscapes [1]. The U.S. National Park Service [27] reorganized the concept of "cultural landscape" and considered that "cultural landscape" is "a lot or area that is associated with historical events, people, activities, or shows traditional aesthetic and cultural values, and contains cultural and natural resources", giving a new connotation to the cultural aspect of the landscape.

*2.3. The Connection between Scene Theory and Cultural Landscape*

The cultural landscape reveals specific cultural connotations through various scenes, and scenes become an important carrier of cultural expression. People understand and give things cultural connotations through scenes, and make things "live". As shown in Figure 2, various types of cultural landscape form amenities, amenity form unique cultural scenes, and cultural scenes are a composite of various amenities, representing the overall cultural

style or aesthetic characteristics of a place. These specific cultural value orientations attract the corresponding cultural subjects to come to cultural practices and form value-emotion judgments and even cultural identity through their feelings.

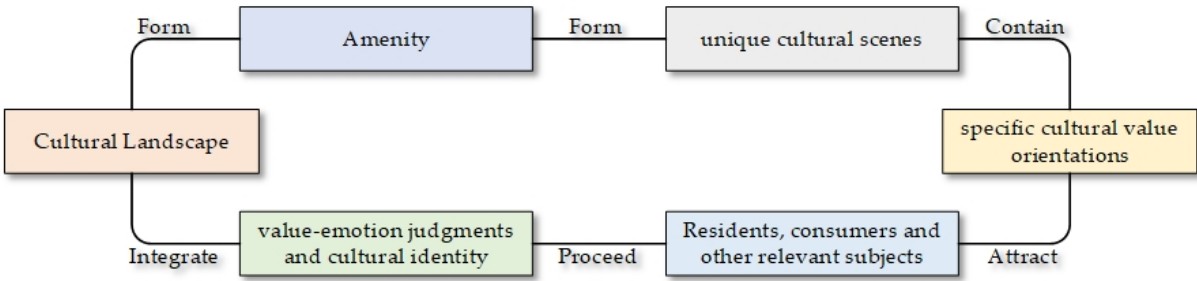

**Figure 2.** The connection between scene theory and cultural landscape.

## 3. Methodology

### 3.1. Overall Framework

The overall framework of cultural landscape reproduction based on scene theory is shown in Figure 3. Based on the scene theory, we first analyze the scene constructed by the cultural landscape and the emotional value dimension embodied in the scene through the collection of online text data and the corresponding data obtained from the questionnaire research, starting from the five elements constructed by the scene theory. Through the deep excavation of the cultural landscape and the historical lineage behind it, the theoretical framework of cultural landscape reproduction of "node-neighborhood-city" is formed, in which "node" is the micro-level expression, mainly portraying the geometric form and attribute characteristics of the elements. "Neighborhood" is a meso-level expression, mainly referring to the expression of the overall appearance of the cultural landscape and the surrounding environment. "City" is a macro-level expression, mainly referring to the expression of the spatial location and distribution pattern of the elements of the cultural landscape, as well as the topography and landscape. Through data collection and processing, we construct a landscape information storage database and carry out visualization and creative expression of the cultural landscape.

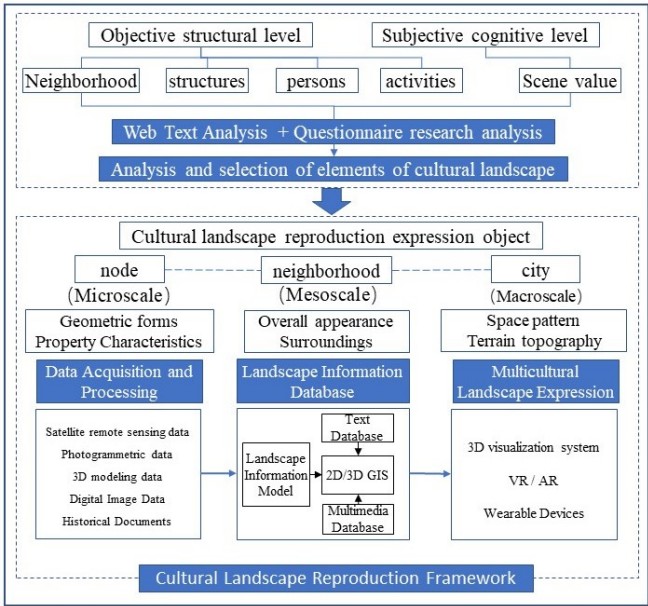

**Figure 3.** The overall framework of cultural landscape reproduction based on scene theory.

*3.2. Cultural Landscape Analysis Based on Scene Theory*

3.2.1. Selection of Analytical Methods

(1)    Content analysis method based on web text

The use of network text analysis methods to analyze cultural landscapes is due to the use of network texts to enrich information collection channels. After entering the information age, a large amount of network evaluation information emerged, and the network gradually expanded the social space for adults to obtain information data, to express their own views, and even had an important practical impact. Network data has the characteristics of high accuracy, wide coverage, large quantity, fast updates, and a large amount of information. Network evaluation information contains both objective perceptions and subjective feelings of people about the city.

Web text analysis method takes the text information of the web as the research object, and realizes high-frequency word analysis, visualization cloud map, semantic network map, and positive and negative affective attitude analysis through knowledge processing and content mining. It provides support for further analysis of cultural landscape scenes.

(2)    Questionnaire analysis method

The questionnaire method is a more widely used method in social surveys at home and abroad. It is a survey method in which the investigator uses a uniformly designed questionnaire to seek information or opinions from the selected respondents. Questionnaire research is the most direct way to understand the evaluation and opinions of the people using the cultural landscape.

The advantages and disadvantages of web text content analysis and questionnaire analysis is shown in Table 1. In this paper, we choose to combine web text analysis and questionnaire analysis, so that the two studies can complement each other, thus making the results of this paper more reliable, more credible, and valid.

**Table 1.** Advantages and disadvantages of web text content analysis and questionnaire analysis.

| Analytical Method | Advantage | Disadvantage |
|---|---|---|
| Content analysis method based on web text | Wide coverage and large numbers Fast updates and great information Non-contact research | Not well targeted Weak in dealing with concepts such as ideology and perception Unable to analyze content that has not been recorded |
| Questionnaire analysis method | Objectivity The results facilitate statistical processing and analysis | Single research method and difficulty to obtain data Questionnaire recall rate is difficult to guarantee |

3.2.2. Data Acquisition and Processing

In the process of acquiring web text data, we first collect web text information related to "Qingjiangpu" on tourism websites through web crawler technology. Then, organize in Word, and delete unnecessary information such as pictures, artificial emoticons, link URLs, and numbers. Finally, it is converted into TXT text format and imported into the ROST CM6 software for word frequency statistical analysis.

Use the ROST CM 6 software to analyze high-frequency words and the semantic network of travel notes. Before analysis, place names, landscape names and proper nouns shall be included in the user-defined vocabulary. Use the ROST CM 6 software to conduct word segmentation and word frequency statistics on TXT documents, obtain preliminary results, and dig out high-frequency words that tourists recognize about Qingjiangpu Ancient City.

### 3.2.3. Objective Structural Level Analysis

(1)    Community and facility analysis

The community is the most appropriate unit for constructing scenarios, and although scenarios can be identified at multiple levels, more emphasis is placed on the community level rather than the city or country level because identifying scenarios at the community level captures the differences within and between those larger units.

(2)    Basic situation analysis of the population

Scene theory suggests that crowds play an important role in the development of cities and regions. Different crowds bring different forces to a city or neighborhood. Any scene will involve different organizations or groups of people, such as residents, tourists, various management departments, artist groups, various organizations of profit or non-profit, etc. The gathering of different groups of people will form different areas, and the characteristics, resource allocation, and social effects they produce will be different.

(3)    Analysis of activity elements

In scene theory, activities link multiple elements such as neighborhoods, facilities, and people. The type and content of activities can reflect the neighborhood environment and cultural connotations side-by-side.

### 3.2.4. Subjective Cognitive Level Analysis

The analysis of the subjective level of perception is mainly measured by the value dimensions of scene theory, which include 15 dimensions. The use of value dimensions can make the originally abstract scene value measurable and can more scientifically describe the cultural value tendencies embedded in specific scenes. Scene value is presented through certain concrete elements, such as facilities and activities. This section uses the form of a semantic differential scale for the analysis of value dimensions.

The semantic difference scale was first developed by C.E. Osgood in a psychometric test, which allows respondents to make judgments on two adjectives with relative meanings according to their subjective feelings on a scale between each pair of adjectives, and the scale's evaluation scales are set with different scores that can be analyzed to compare their evaluation tendencies in turn. The semantic difference scale in this study has 15 words, i.e., 15 sub-dimensions for the scene theory, and five scores are set as −2, −1, 0, 1, and 2, as shown in Table 2.

**Table 2.** Semantic evaluation scale.

| Meaning | Score |
| --- | --- |
| Strongly disagree | L1 (−2 points) |
| disagree | L2 (−1 point) |
| Both are available | L3 (0 points) |
| agree | L4 (1 point) |
| Strongly agree | L5 (2 points) |

### 3.3. The "Node-Neighborhood-City" Approach to Cultural Landscape Recreation

The earth's surface is complex and varied in the objective geographic world in which people live, and even when the same geographic object is observed, people's perspectives or the distance of sight will differ due to variations in visual attention and points of interest, which in turn will produce different visual effects [28]. Therefore, the scene details cannot be described completely and in detail in a single scale, it is necessary to integrate the features of each granularity in the 3D scene from the perspective of multi-level perception to achieve the description and visualization of the 3D scene at different levels.

Aiming at the cultural landscape elements, the model of "node-neighbor-city" is constructed. Among them, "node" refers to the point of great significance and value in the regional cultural landscape, the cultural landscape that persists and plays a key role in the

process of historical development, such as palace sites, sites of important historical events, historical and cultural landmarks, etc. "Node" represents the primary stage of people's interpretation of urban cultural landscape scenes, i.e., fragmented, and unsystematic understanding, and is a micro-level expression. "Neighborhood" refers to the surrounding environment of the node, including the spatial environment and social environment in which the node is located, such as streets and alleys, old trees and trees, patterns, etc. "Neighborhood" is a meso-level expression, mainly referring to the overall appearance of the cultural landscape and its surrounding environment. "Urban" refers to the network connection of nodes based on certain criteria, to form a cultural landscape network and enhance the systemic and coherent nature of the cultural landscape. The criteria for forming a network include, but are not limited to: (i) temporal relevance, cultural landscapes with a strong connection on a temporal scale; (ii) spatial relevance, cultural landscapes with a strong connection on a spatial scale, such as the important cultural landscapes along the linear space of the Beijing-Hangzhou Grand Canal; and (iii) cultural relevance, cultural landscapes expressing the same cultural connotation. "City" is a macro-level expression, mainly referring to the spatial location and the distribution pattern of the elements of the expressive cultural landscape, as well as the display of topographical features and landscape patterns.

In the process of cultural landscape reproduction, data collection and processing are carried out first, including satellite remote sensing data, photogrammetry data, 3D modeling data, digital image data, and historical literature. Among them, satellite remote sensing data is obtained on Google Earth, and digital image data is obtained through online downloading and field shooting. Based on these data, we build 3D models of cultural heritage landscapes through the SketchUp platform. Then, a landscape information storage database is established, and 2D/3D GIS is constructed by integrating a landscape information model, text database, and multimedia database. Finally, visualization and creative expression of cultural landscape scenes are carried out with the help of 2D-3D visualization systems, virtual/augmented reality, and wearable devices.

## 4. Experimental Analysis

### 4.1. Study Area Description

Huai'an, with a history of more than 2200 years, is known as one of the "Four Great Cities" along the canal, along with Suzhou, Hangzhou, and Yangzhou, and is known as the "Canal Capital of China". The ancient city of Qingjiangpu, located on the south bank of the ancient Huaihe River in Huai'an, was once the hub of canal transportation and the key to salt transportation, and was home to the Governor's Office of Canal Transportation and the Jiangnan River Governor's Office, which has a history of over 600 years. Its special climate and geography have made Qingjiangpu a place where people, goods, finance, and culture from the north and south of China meet. It is rich in natural and human landscapes, and it is important to reproduce its cultural landscape.

### 4.2. Selection and Analysis of Cultural Landscape Based on Scene Theory

Based on scene theory, and starting from the five elements included in scene theory, by capturing and interpreting the word frequencies of online texts and field research materials, the landscapes with the most cultural characteristics of cities or communities were selected, and their cultural value dimensions were evaluated.

Firstly, the reviews and pictures about the cultural landscape of Qingjiangpu on tourism network platforms, such as mango.cn, di-anping.com, Ctrip.com, and qunar.com were collected, and about 40,000 words of textual information in the past year were collected for analyzing the scenes constructed by Qingjiangpu, as perceived by tourists. Secondly, the ROST CM6 software was used to analyze the word frequency of the summarized textual materials, and the top 40 high-frequency words related to the research topic were selected according to the frequency of word occurrence. Additionally, the high-frequency words

were recategorized using the five elements of scene theory as the perceptual dimensions (as shown in Table 3).

**Table 3.** Analysis table of the top 40 high-frequency words.

| Perceptual Dimension | Sort | Words | Word Frequency | Perceptual Dimension | Sort | Words | Word Frequency |
|---|---|---|---|---|---|---|---|
| Neighborhood/Community | 2 | Canal | 275 | persons | 29 | Traders | 97 |
| | 4 | Ancient Huaihe River | 261 | | 31 | Craftsman | 76 |
| | 10 | Huajie Street | 226 | | 37 | photographer | 42 |
| | 11 | Dutianmiao Street | 208 | | 39 | inhabitant | 39 |
| | 27 | Dongxi Street | 102 | activities | 7 | Intangible Cultural Heritage Exhibition | 243 |
| structures | 1 | Qingjiang Gate | 293 | | 8 | Temple Fairs | 232 |
| | 3 | Renci Hospital | 274 | | 18 | Stroll | 175 |
| | 6 | Gospel Hall | 251 | | 20 | Photography exhibition | 166 |
| | 9 | Confucious Temple | 231 | | 26 | lion dance | 105 |
| | 13 | Dutian Temple | 207 | | 32 | Canal Cultural Tourism | 76 |
| | 15 | Docks | 187 | | 34 | Photograph | 72 |
| | 17 | Ciyun Temple | 180 | Scene value | 12 | Traditional | 208 |
| | 21 | Chen Pan Ancestral Hall | 144 | | 14 | Comfortable | 206 |
| | 23 | Ancient Mosque | 135 | | 16 | Canal culture | 182 |
| | 24 | Fengji warehouse | 118 | | 19 | happy | 171 |
| | 25 | Zhou Xinfang's former residence | 106 | other | 5 | Southern boats and Northern horses | 260 |
| | 28 | Former Residence of Pearl S. Buck | 98 | | 22 | Huaiyang cuisine | 138 |
| | 30 | Doumu Palace | 80 | | 33 | Straw | 76 |
| | 35 | Shi Dock | 62 | | 36 | Scenic spot | 48 |
| | 40 | City Gate | 37 | | 38 | Sugar | 41 |

Then, field research was conducted in the ancient city of Qingjiangpu, mainly to observe the spatial layout of the cultural landscape, the distribution and characteristics of ancient buildings, the water system, community residents, and the special activities held. Additionally, 30 valid questionnaires were obtained using a survey questionnaire, to examine the feelings of tourists and residents about the city culture conveyed by the cultural landscape scenes, and the evaluation results are shown in Table 4. Through the analysis of high-frequency words and evaluation scales, it can be seen that the cultural landscape of the ancient city of Qingjiangpu has the following characteristics: ① There are many religious architectural and cultural landscapes, and they are all scattered near the Qingjiang Gate, forming a cluster of religious architectural and cultural landscapes. For example, the Gospel Hall, the Confucious Temple, the Durian Temple, the Ciyun Temple, and the Mosque, etc. ② The value dimension of the scenes shown in the ancient city of Qingjiangpu is characterized as local, ethnic, traditional, and friendly (as shown in Figure 4).

Through the analysis of the cultural landscape and historical lineage of the ancient city of Qingjiangpu, we can see that due to some special geographical and human factors, a peculiar landscape has emerged in the ancient city of Qingjiangpu—"South boats and North horses, five religions architecture gathering". Under the condition of "south boats and north horses", the officials and merchants from the south to the north had frequent economic and cultural exchanges, and also brought various faiths to Qingjiangpu, forming a multi-religious cultural center. The five major religions of Confucianism, Buddhism, Taoism, Islam, and Christianity are gathered in the vicinity of Qingjiang Gate, forming a unique religious and cultural landscape that demonstrates the openness and inclusiveness of Qingjiangpu. In the following, we will collect historical documents, satellite remote sensing, 3D modeling, and other data to build a database for storing landscape information, and present the unique architectural and cultural landscape of "Southern boats and Northern horses, five religions architecture gathering" from multiple dimensions and scales.

**Table 4.** Semantic rating scale.

| Variable Content | Average of Scores | The Proportion of Frequency Distribution of the Data (%) | | | | |
|---|---|---|---|---|---|---|
| | | L1 Strongly Disagree (−2) | L2 Disagree (−1) | L3 Both Are Available (0) | L4 Agree (1) | L5 Strongly Agree (2) |
| Local | 0.87 | 3.33% | 6.67% | 13.33% | 53.33% | 23.33% |
| Ethnic | 0.87 | 3.33% | 13.33% | 16.67% | 26.67% | 40.00% |
| Corporate | −0.20 | 13.33% | 26.67% | 33.33% | 20.00% | 6.67% |
| State | 0.80 | 0.00% | 13.33% | 20.00% | 40.00% | 26.67% |
| Rational | −0.13 | 10.00% | 16.67% | 53.33% | 16.67% | 3.33% |
| Traditional | 0.60 | 3.33% | 6.67% | 36.67% | 33.33% | 20.00% |
| Self-expressive | 0.07 | 10.00% | 20.00% | 26.67% | 40.00% | 3.33% |
| Utilitarian | −0.10 | 0.00% | 33.33% | 43.33% | 23.33% | 0.00% |
| Egalitarian | 0.63 | 0.00% | 3.33% | 33.33% | 60.00% | 3.33% |
| Glamorous | 0.73 | 3.33% | 10.00% | 20.00% | 43.33% | 23.33% |
| Glamorous | 0.10 | 10.00% | 23.33% | 23.33% | 33.33% | 10.00% |
| Formal | −0.83 | 33.33% | 26.67% | 30.00% | 10.00% | 0.00% |
| Neighborly | 1.13 | 0.00% | 0.00% | 10.00% | 66.67% | 23.33% |
| Transgressive | 0.07 | 3.33% | 23.33% | 40.00% | 30.00% | 3.33% |
| Exhibitionistic | −0.37 | 30.00% | 10.00% | 36.67% | 13.33% | 10.00% |

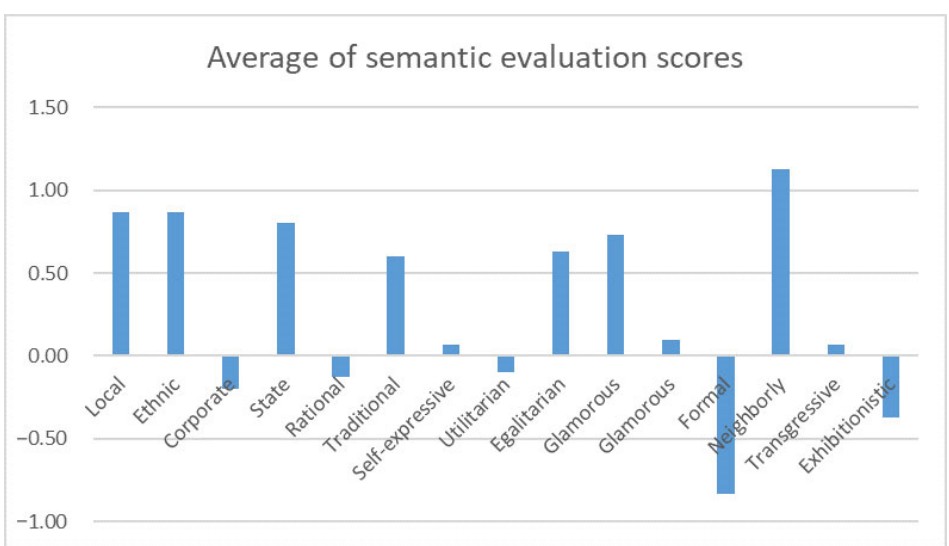

**Figure 4.** Mean semantic evaluation graph.

### 4.3. Qingjiangpu Religious Architecture Cultural Landscape Reproduction

Based on the theoretical framework of "node-neighborhood-city", firstly, at the macroscopic scale, the geographical environment of the formation of the cultural landscape of religious architecture in Qingjiangpu and the canal landscape of "southern boats and northern horses" is shown through 2D maps; at the mesoscopic scale, the cultural landscape of the ancient city of Qingjiangpu, the hub of the Beijing-Hangzhou Grand Canal, is shown through the combination of 2D maps and 3D models; at the microscopic scale, a virtual scene visualization system based on Unity 3D is developed to further show the details of the 3D scenes of religious architecture.

#### 4.3.1. Macroscale

To show the geographical environment factors of the development and evolution of the typical religious and cultural landscape of the ancient city of Qingjiangpu, the history of the formation of the canal landscape of "South Boats and North Horses" is shown on the map, as shown in Figure 5.

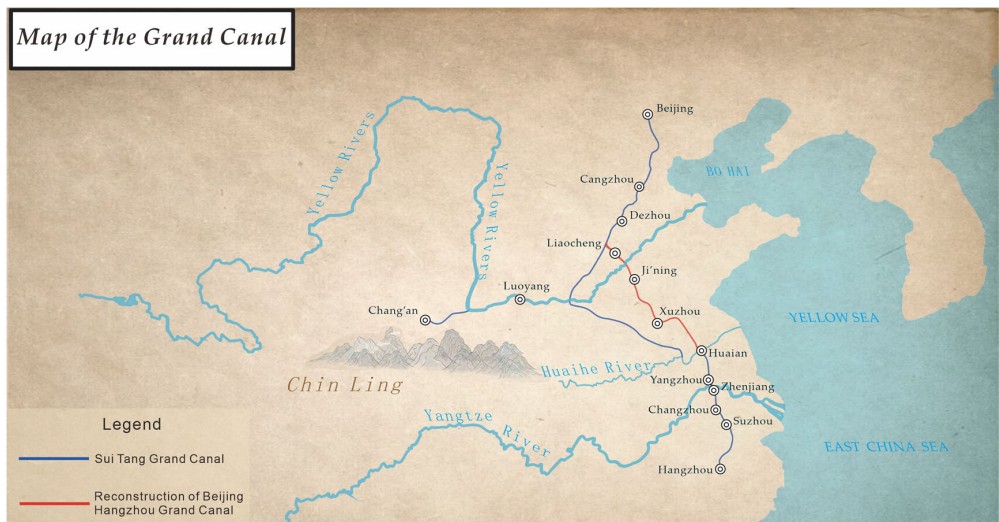

**Figure 5.** Map of the Grand Canal.

### 4.3.2. Mesoscale

The integration of 2D maps and 3D spatial scenes is an important application of GIS. In the existing technical solutions, the 3D model is mainly used for spatial display, showing the visual effect of the spatial form and color appearance of the object. 2D maps are used to show the layout of the city. In practical application, the advantages of 2D GIS and 3D models are combined to achieve integrated expression. The overall appearance of the cultural landscape and the surrounding environment are presented using a 3D model and 2D remote sensing map overlay, showing the cultural landscape of "five religions converging" in the Qingjiangpu area in the context of "South Boat and North Horse". The scenes visually show the natural and humanistic elements such as water systems, vegetation, roads, and residential areas. It is shown in Figure 6.

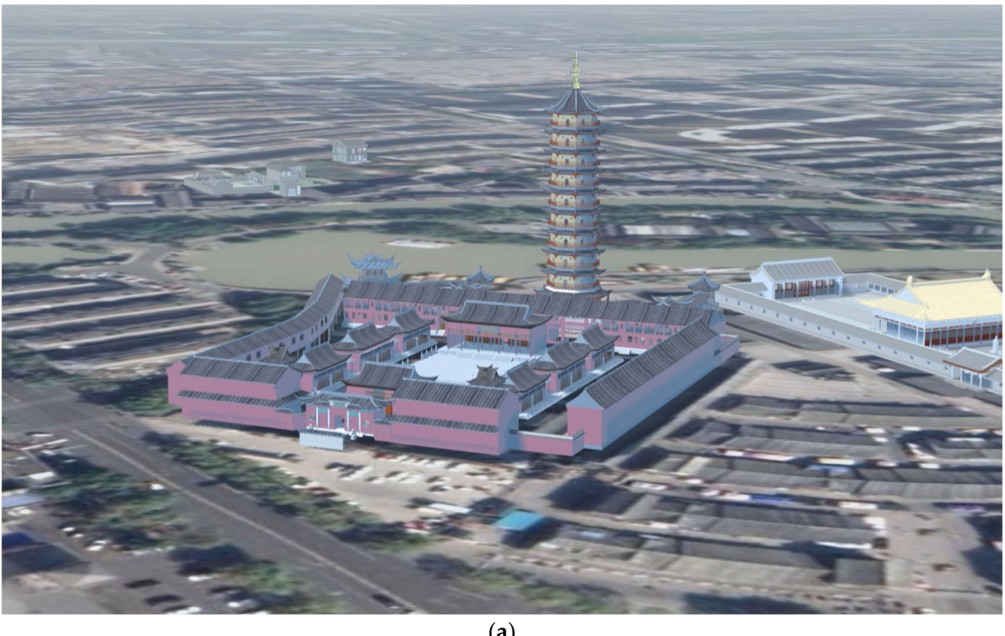

(**a**)

**Figure 6.** *Cont.*

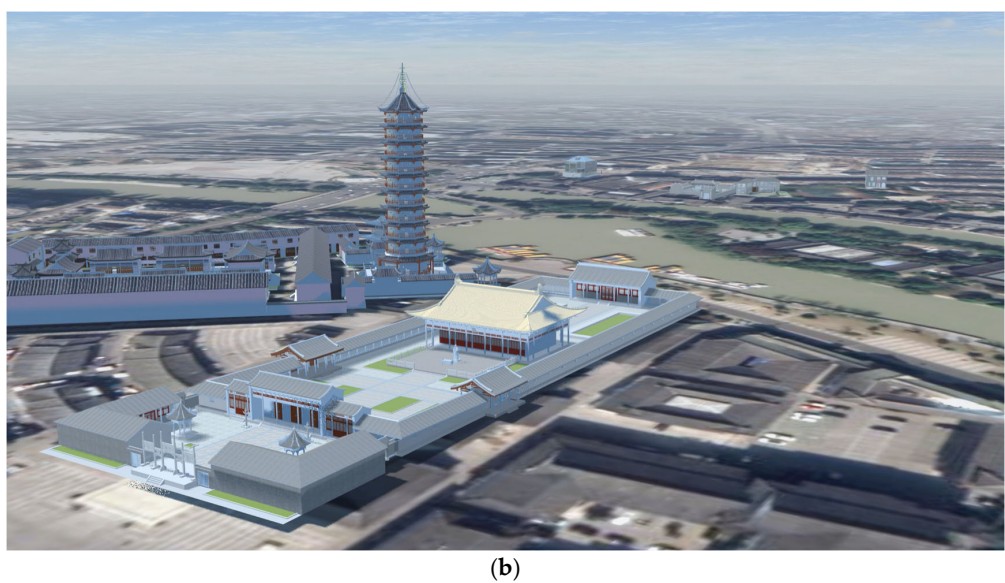

(**b**)

**Figure 6.** Two-three dimensional fusion scene realization. (**a**) Ciyun Temple, (**b**) Confucious Temple.

4.3.3. Microscale

　　To show the cultural characteristics of the five major religious buildings in Qingjiangpu, this study is based on Unity 3D and C# programming language for system development, reproducing the geometric form and attribute characteristics of the religious building model of Qingjiangpu in full detail. The specific workflow was mainly divided into three parts: three-dimensional modeling, model integration, and system function realization, as shown in the Figure 7.

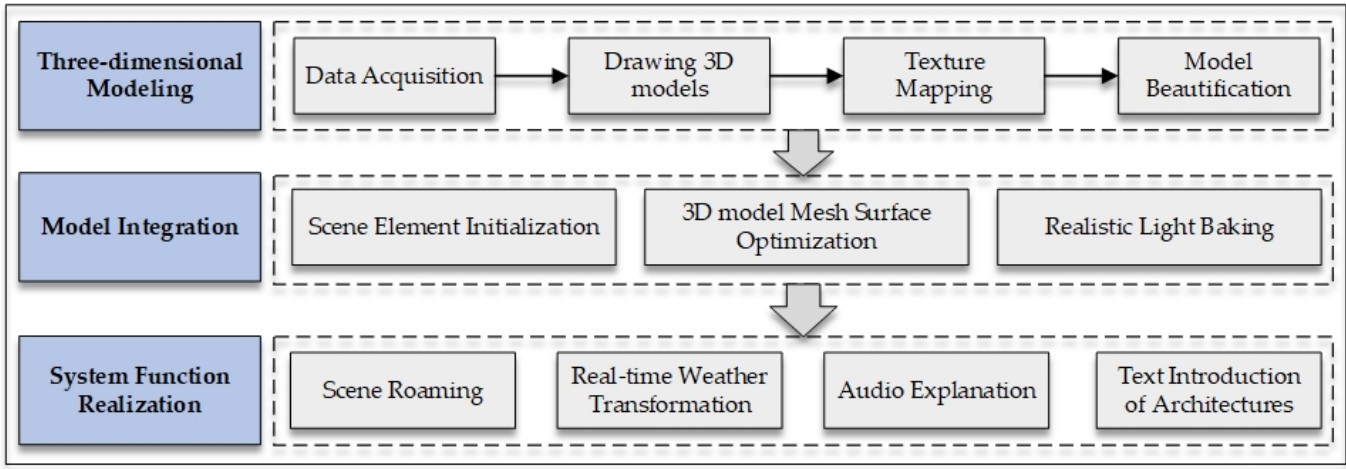

**Figure 7.** Cultural landscape 3D model construction and visualization workflow.

(1)　Three-dimensional modeling

　　First, we collected historical documents, remote sensing images, mapping drawings, and high-definition images of the cultural heritage landscape of the ancient city of Qingjiangpu. Secondly, based on these data, the preliminary 3D models were constructed using the SketchUp modeling software. Third, the models were textured and mapped using high-definition images. Finally, model beautification was performed to make it look more realistic.

(2)    Model integration

Scene element initialization: There are several model import methods in Unity 3D, and this study uses drag-and-drop to import the 3D model built in step (1). After importing, select the model file and set its material import method to Use External Materials (Legacy). In order to get a better light baking effect, it is necessary to check its Calculate Normals module when importing the model, and check the option to automatically generate Lightmap UVs. Considering the problem of inconsistent unit scale between Unity and SketchUp, set the model Scale to 0.001 in Unity 3D.

3D model mesh surface optimization: On the window interface, classify and combine the 3D model files, and use the integrated Simple LOD plug-in to simplify and optimize the mesh surface of the sub-objects in the combination, so as to reduce the draw call value and avoid the problem of scene jam when the program is loaded.

Realistic light baking: Select the model after mesh surface optimization, set it as Static object, and select the scene lighting object, set its rendering mode to Baked, in the Lighting component, check Baked Lightmap, and set the baking option to Shadow Mask, and keep the rest as default parameters, and perform scene lighting baking to get a realistic scene lighting rendering effect.

(3)    System function realization

This study mainly reproduces the scenes of religious buildings in Qingjiangpu and shows the cultural characteristics and connotations of religious buildings, and realizes the following functions: full-scene roaming of religious buildings, real-time weather change, voice explanation, and architectural text introduction; the system interface is shown in Figure 8.

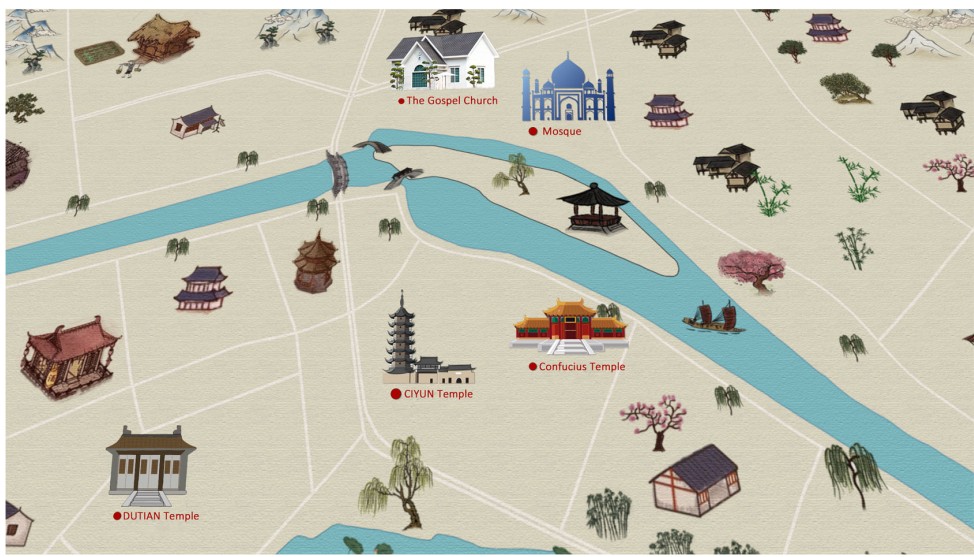

**Figure 8.** Initial interface of religious scene reproduction system based on a micro-cosmic scale.

As shown in Figure 9, the system scene roaming page, the right menu bar has six parts: start roaming, pause roaming, continue roaming, voice explanation, system settings (sound), and return to the map; the small map in the lower left corner of the interface shows its location within the scene; the bottom of the interface shows the introduction of the architectural and cultural landscape when roaming.

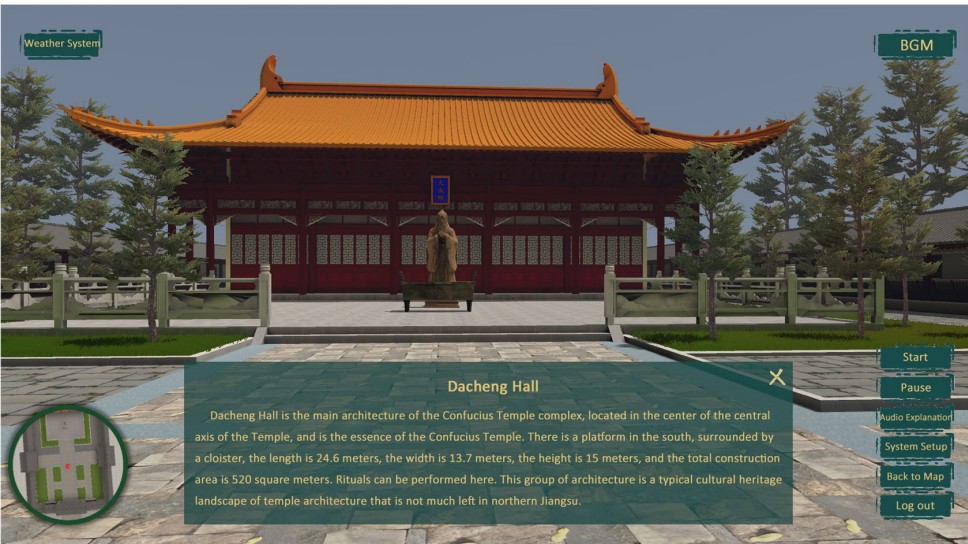

**Figure 9.** Roaming interface of religious scene reproduction system based on a microcosmic scale.

*4.4. Result Analysis and Discussion*

Most of the current research starts from a technological perspective, including digital modeling, digital presentation, and human-computer interaction. This paper focuses on these technologies and emphasizes the excavation of the cultural connotation itself. Compared with traditional methods, the proposed method has the following two advantages. (1) Based on scene theory, scientifically select and determine the theme content of urban cultural landscape reproduction. The concept of "scene" is used to explain the cultural connotation in the space, to analyze the local cultural style, and to find the cultural characteristics of the place. Moreover, the value evaluation system proposed by the scene theory provides a reference basis for us to quantify the selected elements. (2) Based on the theoretical framework of "node-neighborhood-city" cultural landscape reproduction, it shows the connotation of the urban historical and cultural landscape from multiple dimensions and scales.

The method proposed in this paper is innovative in the existing research of cultural heritage landscape visualization, but there are still some deficiencies, which are expected to be completed in the next research. (1) In the analysis of subjective cognition, this paper used three main dimensions and 15 sub-dimensions related to value evaluation in the scene theory to analyze the cultural characteristics of Qingjiangpu Ancient City. In future research, the analysis method of scene theory can be applied to design a scene value evaluation index system that meets the characteristics of Chinese cultural heritage landscape, so that the research results can be more objective and reasonable. (2) The scene theory referred to in this paper is the "scene" of the New Chicago School in the field of sociological research. In subsequent research, the theories and methods of virtual geographic environment (VGE) and geographic scenario [29,30] can be combined to carry out studies such as virtual online tourism and the holographic display of cultural heritage landscapes.

**5. Conclusions**

The reproduction of the cultural landscape can reproduce the cultural connotation it conveys, which is of great significance to the development of the city and the inheritance of culture. First of all, the reproduction of the religious cultural landscape of the ancient city of Qingjiangpu will, on the one hand, show the culture of Qingjiangpu in a more intuitive way, which is conducive to the dissemination of the cultural connotation of Qingjiangpu, and thus promote the development of the cultural industry of Qingjiangpu. Secondly, under the joint interweaving of traditional canal culture and western foreign culture, the religious architectural cultural landscape with unique oriental characteristics has developed and evolved in the ancient city of Qingjiangpu. The reproduction of the religious cultural

landscape of the ancient city is undoubtedly a compliment and improvement to the typical religious landscape scenes of the world. This paper proposes a theoretical framework of "node-neighborhood-city" by analyzing the cultural landscape scenes and cultural values and conducting a deep excavation of the cultural landscape and the historical lineage of its formation based on scene theory. The cultural landscape of the ancient city of Qingjiangpu, with its unique religious architecture, is reproduced from three different scales: macro, meso, and micro. This study will provide a theoretical basis and practical reference for the study of cultural landscape restoration.

**Author Contributions:** Conceptualization, W.M., J.S. (Junchao Shen) and J.S. (Jie Shen); methodology, W.M. and J.S. (Jie Shen); software, S.H., T.C. and J.S. (Junchao Shen); validation, J.S. (Jie Shen), T.C. and J.S. (Junchao Shen); investigation, W.M., S.H., J.S. (Jie Shen), J.S. (Junchao Shen) and T.C.; writing—original draft preparation, W.M. and T.C.; writing—review and editing, J.S. (Jie Shen); project administration, J.S. (Jie Shen); funding acquisition, J.S. (Jie Shen). All authors have read and agreed to the published version of the manuscript.

**Funding:** This research was funded by The National Natural Science Foundation of China, grant number 41930104, and National Key R&D Program of China, grant number 2021YFE0112300.

**Institutional Review Board Statement:** Not applicable.

**Informed Consent Statement:** Not applicable.

**Data Availability Statement:** Not applicable.

**Acknowledgments:** The authors are grateful to the Huai'an Natural Resources and Planning Bureau, Huai'an Archives, Huai'an Huaizhong Alumni Science and Technology Innovation Alliance, and the staff of Huai'an Academy of Cultural Relics Protection and Archaeology for providing a large amount of data. Thank Visiontek INC (Nanjing) for providing 3D modeling technical guidance. We thank Ying Li, Jiale Li, and Zhiyu Wu of Jiangsu Second Normal University, as well as Shiwei Li, Li Bei, Shenpeng Qiu, and Haoyu Yang of Nanjing Normal University for participating in the 3D modeling and page design in this experiment. Thanks to Nina Sun for her valuable comments during the writing process of the paper.

**Conflicts of Interest:** The authors declare no conflict of interest.

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
