# Peer review of "Cultural Landscape Reproduction of Typical Religious Architecture in Qingjiangpu Based on Scene Theory"

_applsci, doi:10.3390/app13010082_

Round 1

Reviewer 1 Report

The paper is well structured and it presents an interesting and original research, based on the elaboration of known applications and theories in a creative and impactfull way.

It is suggested to the authors to extend and detail the state of the art in the introduction section. References are presented in a generical way, and they do not specify the valuable meaning assumed by the authors to support their specific research idea.

Lines 260-263 contain a typo of syntax.

The methodology section must be detailed with more technical information about the applied methodological steps, platforms, and data processing.

Author Response

Dear reviewer,

Thanks for your comments concerning our manuscript entitled “Cultural Landscape Reproduction of Typical Religious Architecture in Qingjiangpu Based on Scene Theory” (Manuscript ID: applsci-2034238). Those comments are valuable and very helpful. We have read through the comments carefully and made corrections based on your comments. We resubmit the revised manuscript. We highly appreciate your time and consideration. Major revisions are marked in red. 

Reviewer 2 Report

The paper is interesting but needs some improvements:

1. Lines 34-41: the sentence is too long and his difficult to comprehend

2. Lines 49-67: This list is fine but incomplete: if you talk about photogrammetry and laser scanning for the survey of Cultural Heritage and archaeological sites you have also to check works of:

- Fabio Remondino

- Gabriele Guidi

- Pierre Grussenmeyer

- Diego Gonzalez Aguilera

- John Mills

as an example

3. Lines 112-117: This sentence need to be written better, the meaning of it is not so easily understandable and the construction is not clear.

4. Lines 126-131: explain better

5. Lines 172-175: check the verb tense

6. Line 185: Please re write the sentence which meaning is too simple and obvious

7. Lines 188-192: Please check this sentence and explain better. maybe an explanatory table can help

8. Lines 227-230: explain better and check the sentence

9. Lined 339 you state "3D model and 2D remote sensing map overlay". Why? This sentence needs to be better explained

10. Line 354: you talk about 3D model. Where did you find this model? or, if you did it, how did you produce the model? which software did you use? On which archival or survey data have you based the reconstruction of this temple?

11. Lines 359-362: why is this passage needed?

12. Lines 395-402: explain better

13. References need update.

Thank you!

Author Response

(The authors gave the same response as above.)
